# Modulating NO–GC Pathway in Pulmonary Arterial Hypertension

**DOI:** 10.3390/ijms25010036

**Published:** 2023-12-19

**Authors:** Anna D’Agostino, Lorena Gioia Lanzafame, Lorena Buono, Giulia Crisci, Roberta D’Assante, Ilaria Leone, Luigi De Vito, Eduardo Bossone, Antonio Cittadini, Alberto Maria Marra

**Affiliations:** 1IRCCS SYNLAB SDN, Via Emanuele Gianturco 113, 80143 Naples, Italy; anna.dagostino@synlab.it (A.D.); lorena.buono@szn.it (L.B.); ilaria.leone@synlab.it (I.L.); 2Department of Clinical and Experimental Medicine, Internal Medicine, Garibaldi Hospital, University of Catania, Via Palermo 636, 95122 Catania, Italy; lorenalanzafame@gmail.com; 3Department of Translational Medical Sciences, “Federico II” University of Naples, Via Pansini 5, 80131 Naples, Italy; giulia.crisci@unina.it (G.C.); roberta.dassante@outlook.it (R.D.); luisdevito@inwind.it (L.D.V.); antonio.cittadini@unina.it (A.C.); 4Department of Public Health, “Federico II” University of Naples, Via Pansini 5, 80131 Naples, Italy; eduardo.bossone@unina.it; 5Gender Interdipartimental Institute of Research (GENESIS), “Federico II” University of Naples, Via Pansini 5, 80131 Naples, Italy

**Keywords:** pulmonary arterial hypertension, nitric oxide, riociguat, right ventricle

## Abstract

The pathogenesis of complex diseases such as pulmonary arterial hypertension (PAH) is entirely rooted in changes in the expression of some vasoactive factors. These play a significant role in the onset and progression of the disease. Indeed, PAH has been associated with pathophysiologic alterations in vascular function. These are often dictated by increased oxidative stress and impaired modulation of the nitric oxide (NO) pathway. NO reduces the uncontrolled proliferation of vascular smooth muscle cells that leads to occlusion of vessels and an increase in pulmonary vascular resistances, which is the mainstay of PAH development. To date, two classes of NO-pathway modulating drugs are approved for the treatment of PAH: the phosphodiesterase-5 inhibitors (PD5i), sildenafil and tadalafil, and the soluble guanylate cyclase activator (sGC), riociguat. Both drugs provide considerable improvement in exercise capacity and pulmonary hemodynamics. PD5i are the recommended drugs for first-line PAH treatment, whereas sGCs are also the only drug approved for the treatment of resistant or inoperable chronic thromboembolic pulmonary hypertension. In this review, we will focus on the current information regarding the nitric oxide pathway and its modulation in PAH.

## 1. Introduction

Pulmonary hypertension (PH) is a clinical condition defined by the presence of a mean pulmonary arterial pressure (mPAP) higher than 20 mmHg, assessed invasively by right heart catheterization [1,2]. Several conditions can lead to its development and the most common are left heart failure, pulmonary parenchymal diseases, and chronic pulmonary thromboembolism. Regardless of its cause, PH commonly progresses to right heart failure and death. A rare but well-characterized cause of PH is pulmonary arterial hypertension (PAH), which is a chronic and progressive vascular disease of the lung microcirculation whose hallmark is the presence of plexiform lesions [1]. Once considered an orphan disease with an inexorable poor prognosis, PAH is a valuable prototype of a rare and severe disease whose natural course has been transformed by research and clinical example. Indeed, since the first trial of Barst and co-authors using epoprostenol in 1996 [3], nowadays, twelve molecules belonging to three different classes have been approved for PAH treatment. These drugs are able to modulate three different molecular pathways [1]. Chiefly, the nitric oxide (NO) pathway has been extensively investigated. To date, three molecules modulating this pathway have been approved, two belonging to the category of PD5i (sildenafil and tadalafil) for first-line therapy and one to the sGC category (riociguat) for second-line treatment in case of lack of improvement with PD5i [1]. Clearly, the NO pathway plays a pivotal role in PH; therefore, the aim of this narrative review is to elucidate its function and the clinical implication of its modulation.

## 2. Overview of the NO Pathway

Nitric oxide (NO) is a critical component of vasculature and works by activating soluble guanylate cyclase (sGC), which catalyzes the conversion of guanosine triphosphate (GTP) to cyclic guanosine monophosphate (cGMP) [4,5]. As a potent vasodilator, NO is a key signal regulator of numerous physiological functions. Indeed, NO is responsible for blood vessels’ tone, perfusion, and function in all major organs. It demonstrates ubiquitous effects such as modulating synaptic plasticity in the brain, platelet aggregation, skin function, and numerous other physiological processes such as myocardial function. NO fulfils its function by targeting and activating soluble guanylate cyclase (sGC) [6,7,8]. Several early studies played key roles in elucidating the mechanism of the NO–sGC–cGMP signaling pathway. These reports elucidated that endogenous NO is synthetized from arginine by a group of three calmodulin-dependent NO synthase (NOS) enzymes. Two of these NOS enzymes are constitutively expressed, namely endothelial NOS (eNOS) and neuronal NOS (nNOS), whose activities are stimulated by increases in intracellular calcium [9]. The third class of NOS enzymes are those involved in immune reactions and are induced (iNOS) by transcriptional activation mediated by specific cytokine combinations [9]. The reaction of NO synthesis is catalyzed in all cases by a five-electron oxidation of arginine with the consequent formation of NO and citrulline (Figure 1).

## 3. NOS Regulation under Physiological Conditions

eNOS is a 133 kDa protein encoded by the NOS3 gene and it is expressed in endothelial cells (from vascular endothelium and endocardium) but also in cardiac myocytes and platelets [10,11,12]. Arginine bioavailability is the limiting factor of its production. In its absence, monomeric eNOS is unable to supply electrons to the heme center of other eNOS monomers. Therefore, the electrons of the eNOS monomer are transferred to a position where they interact with oxygen, thus facilitating the formation of superoxide [4,13]. Consequently, eNOS monomer dimerization is crucial for proper enzymatic activity and, subsequently, NO production. eNOS regulation is a complex process mainly ruled by phosphorylation and dephosphorylation and one of the principal regulators is heat shock protein 90 (HSP90). Indeed, silencing of this protein seems to sensibly destabilize eNOS dimers. Phosphorylation of the NH_2_-terminal oxygenase domain activates the ability of the eNOS monomer to bind zinc, heme, and tetrahydrobiopterin (BH4) and coordinates eNOS homodimer formation. G-protein coupled receptor (GPCR) kinase-interacting protein-1 (GIT1) tyrosine phosphorylation by Src is what activates eNOS; Akt regulates both Src’s capacity to phosphorylate GIT1 and the interaction between GIT1 and eNOS [4]. Several transcriptional activators have been identified and, among them, there are shear stress and stretch and Krüppel-like factor 2 (KLF2) [14], but also reactive oxygen species (ROS, such as hydrogen peroxide (H_2_O_2_)). Moreover, eNos can also be regulated epigenetically through DNA methylation, and several eNOS polymorphisms influencing gene expression have been identified [15,16]. On the other hand, iNOS is a protein of 130 kDa that is encoded by the NOS2 gene. As mentioned before, iNOS transcription is activated because of a specific cytokine expression pattern that happens in pro-inflammatory conditions, for example, and is expressed in various cell types as leukocytes, endothelial cells, vascular smooth muscle cells (VSMCs), cardiac myocytes, nerve cells, and fibroblasts [17,18,19]. As stated above, iNOS activity is calcium-independent and maintains, therefore, a high NO output until exhaustion of the substrate and cofactors or enzyme degradation [19,20]. Finally, nNOS is encoded by the NOS1 gene. nNOS is abundantly expressed in neurons and many of its functions are associated with the control of neuronal homeostasis. Additionally, it has been demonstrated that nNOS is also expressed in other cell types such as endothelial and smooth muscle cells, thus associating nNOS with an important role in regulating homeostasis in human vasculature [21].

## 4. NOS Downstream Signaling and Its Effects on Cardiovascular System

The principal intracellular target for NO is recognized to be soluble guanylate cyclase (sGC), which converts GTP into the second messenger cyclic guanosine monophosphate (cGMP) [22]. cGMP is responsible for a variety of downstream effects such as leucocyte adhesion, relaxation of the vascular smooth muscle, and inhibition of platelet aggregation [23]. Downstream occurrences in this pathway seem to be highly dependent on the cell type, and the expression of pathway components can also vary considerably depending on the tissue or whether it is physiological or pathological. Briefly, the pathway can be summarized as follows: the conversion of arginine to citrulline and NO activates sGC, whose enzymatic activity converts GTP to cGMP, which is the second messenger [24].

cGMP downstream effects are mediated by several effectors that are commonly known as cGMP-regulated protein kinases (abbreviated as cGKs or PKGs), which are responsible for the phosphorylation of cGMP target protein. This signaling cascade can be disrupted by cGMP hydrolysis via phosphodiesterases (PDEs) or cGMP export [24].

As aforementioned, all three isoforms of NOS have been detected in mammalian cardiomyocytes, vascular endothelial cells, and vascular smooth muscle cells (VSMCs) [12,25]. Moreover, it has been demonstrated that eNOS-synthesized NO induces a slow increase in sarcomere shortening and Ca^2+^ transient amplitude through the enhancement of Ca^2+^ release from the sarcoplasmic reticulum (SR) [26]. Conversely, nNOS seems to be involved in spontaneous diastolic Ca^2+^ sparks in afterload-constrained cardiac myocytes. The NOS enzymes also regulate cardiac myocyte basal contractility and β-adrenergic responsiveness [27]. However, the effects may differ according to the anatomical variations between atria and ventricles, between whole hearts and isolated myocytes, and between species [28]. In the vessel wall, eNOS-mediated NO synthesis promotes relaxation and inhibits the proliferation of VMSC, while, when diffusing in the lumen, it mediates angiogenesis, inhibits platelet aggregation, and, in turn, prevents the occurrence of thrombosis. Synthesis of NO by nNOS within VSMCs also contributes to the regulation of vascular tone [29].

Under physiological conditions, NOS–NO signaling has both autocrine and paracrine effects and mediates cardiovascular function by activating several downstream pathways involved in the regulation of vascular function, hemostasias, and cardiac myocyte function [29]. In particular, in the healthy myocardium, NO activates the sGC–cGMP protein kinase (PKG) pathway and regulates intracellular Ca^2+^ levels in two main ways. The first consists of direct phosphorylation of myosin-binding protein C (MYBPC) and troponin I (TnI), which are responsible for the reduction in myofilament Ca^2+^ sensitivity (Figure 2).

The latter consists of the block of Ca^2+^ influx through the inhibition of the sarcolemma L-type Ca^2+^ channel (LTCC) at the transverse-tubule–SR junction. The reduction in cytosolic Ca^2+^ concentration is also supported and encouraged by NO produced by nNOS through the activation of PKG that phosphorylates phospholamban (PLB), which uncouples from the sarcoplasmic/endoplasmic reticulum calcium-ATPase (SERCA) pump and is free to mediate the reuptake of Ca^2+^ from the cytoplasm to the sarcoplasmic reticulum (SR). Inhibition by nNOS of xanthine oxidoreductase (XOR) also promotes the decrease in sarcomeric protein Ca^2+^ sensitivity [30] (Figure 3). Conditions in which the normal production and/or NO targeting are disrupted are potentially pathologic, and reduced NO generation by NOS typically coexists with its increased synthesis of O_2_^−^, which is a condition referred to as NOS uncoupling [31].

Another aspect that needs to be considered is the expression of iNOS in the myocardium. iNOS is expressed only in tissues stimulated by cytokines. In disease states, the expression of iNOS has been clearly demonstrated in the heart, including cardiac myocytes [32]. Interferon (IFN)-γ and IL-6 have also been shown to induce iNOS mRNA and activity in cardiac myocytes [18,33].

So, the impairment of endothelium-dependent, NO-mediated vasodilator function is a generalized phenomenon in heart failure. As in other pathologies in which endothelial dysfunction has been observed, in heart failure, there is a clear loss in the end endothelial production and/or bioavailability of NO. The reduction in eNOS expression or reduced bioavailability of NO secondary to oxidative inactivation by superoxide (O^−^) are some of the possible leading causes of endothelial dysfunction [34].

## 5. NO–GC Pathway in PAH

The role of NO regulation in PAH is particularly intriguing. All three NOS isoforms are expressed by the pulmonary vasculature but not all are involved in the same way in the pathogenesis of PAH. Fagan et al. demonstrated that only the ablation on eNOS is crucial for the establishment of increased pulmonary pressure in mouse lung. Disruption of iNOS signaling only has a few effects on the pulmonary pressure level, in particular, only very mild elevated PAP, while nNOS disruption did not have any effect [35]. However, the inflammatory state established in PAH may induce iNOS, so its role in pathogenesis cannot be entirely excluded. Nevertheless, dysregulation of eNOS is not the exclusive cause of PAH; indeed, Fagan et al. showed that eNOS knock-out rats had increased levels of iNOS and consequent increased circulating NO, which suggests a compensatory effect [35,36]. The whole picture is complicated by the presence of concomitant factors that increase and decrease eNOS expression in PAH. Furthermore, endothelial cell lesions and plexiform regions tend to be topically located in the pulmonary arteries in PAH, contributing to focal discrepancies in eNOS expression. The mechanisms involved in eNOS regulation in PAH include growth factors, endothelin-1 and serotonin. In addition, pulmonary blood flow is sped up in PAH and this leads to stress, which is a potent inducer for eNOS [37,38,39,40]. Vascular endothelial growth factor (VEGF) can also induce the production of NO, which is mainly produced in PAH plexiform lesions and participates in vascular remodeling [37]. Indeed, the capability of NO to reduce the uncontrolled proliferation of vascular smooth muscle cells that lead to the occlusion of vessels and an increase in pulmonary vascular resistances (more than a mere vasodilation) is believed to be the main mechanism of action of PD5i and sGC in PAH.

## 6. Modulation of Pulmonary Vascular NO in PAH

Three major molecular pathways have been demonstrated to be involved in the development of PAH and are the target for twelve molecules: the endothelin pathway, the prostacyclin pathway, and the nitric oxide/cyclic guanosine monophosphate (cGMP) pathway.

To date, two classes of drugs modulating the NO/cGMP pathway have been approved in pulmonary arterial hypertension treatment: phosphodiesterase type 5 (PDE5) inhibitors and the soluble guanylate cyclase stimulator (sGC).

### 6.1. Phosphodiesterase Type 5 Inhibitors

Three PDE5 inhibitors have been mainly evaluated in patients with PAH: sildenafil, tadalafil, and vardenafil. Several uncontrolled studies have reported the positive effects of sildenafil in once so-called primary pulmonary hypertension [41,42,43]. In 2004, Sastry et al. performed the first controlled trial of sildenafil and compared it with a placebo [43,44]. In this randomized double-blind study, 22 patients with idiopathic pulmonary arterial hypertension, in class II–III of NYHA, were randomized, and 12 of them received placebo and 10 sildenafil. Patients included were evaluated with Doppler echocardiography and a treadmill exercise test after six weeks; at the end of this period, they crossed over to alternate therapy for other six weeks. The authors considered a change in exercise time on the treadmill using the Naughton protocol as the primary endpoint. The results of the study showed a 44% increase in the mean from 475 ± 168 s at the end of the placebo phase to 686 ± 224 s at the end of the sildenafil phase (*p* < 0.0001). Regarding the secondary endpoints, the cardiac index was significantly improved from 2.80 ± 0.90 L/m^2^ at the end of the placebo phase to 3.45 ± 1.16 L/m^2^ at the end of six weeks of sildenafil therapy (*p* < 0.0001); similar beneficial results were observed in clinical outcomes regarding dyspnea and fatigue. In contrast, a significant change in pulmonary artery systolic pressure was not seen between both phases.

The effects of different dosages of sildenafil were evaluated in SUPER-I (Sildenafil Use in Pulmonary Arterial Hypertension) [45]. This placebo-controlled clinical trial included 277 patients; all of the study population was randomized in four treatment groups: three groups received sildenafil at different dosages of 20 mg (*n* = 69), 40 mg (*n* = 67), or 80 mg (*n* = 71), and the last group was placebo (*n* = 70). After 12 weeks, exercise capacity was assessed by the mean of six-minutes walking distance (6MWD) and the times improved in all subgroups receiving sildenafil; specifically, the improvement was better in the group with a higher dose of sildenafil (+13% in 20 mg group, +13.3% in 40 mg group and +14.7% in 80 mg group) (*p* < 0.001, for all comparisons). Furthermore, sildenafil significantly improved the cardiopulmonary hemodynamics and World Health Organization (WHO) functional class of the groups after 12 weeks, as compared with changes in the placebo group.

Further studies compared adding sildenafil to others classes of drugs indicated for use in PAH treatment to evaluate a combination therapy. In particular, intravenous epoprostenol, a prostacyclin analogue, was used before sildenafil in pulmonary arterial hypertension. In this regard, the Pulmonary Arterial Hypertension Combination Study of Epoprostenol and Sildenafil (PACES study) showed some favorable effects of adding sildenafil to epoprostenol [46] in a study population with pulmonary arterial hypertension in long-term treatment with intravenous epoprostenol. All of the 264 patients were randomly assigned to a placebo group (*n* = 133) or sildenafil group (*n* = 134). At the end of 16 weeks, the authors analyzed the change in exercise capacity measured by 6MWD and defined the primary outcome of the study between patients receiving sildenafil plus epoprostenol and epoprostenol monotherapy. The increase in distance was about 30 m for the sildenafil group (*p* < 0.001 vs. placebo). Furthermore, the addition of sildenafil had positive effects on hemodynamic evaluation compared with epoprostenol alone. In fact, the combination therapy improved mean pulmonary arterial pressure, systemic vascular resistance, pulmonary vascular resistance, and cardiac output, as well as lengthening the time to clinical worsening. A relevant result was that patients with higher walking capacity at baseline had the most benefit in exercise capacity using combination therapy.

Tadalafil is another treatment option as a phosphodiesterase type 5 inhibitor. The aim of the PHIRST study (Pulmonary Arterial Hypertension and Response to Tadalafil) was to evaluate the effects of tadalafil in 405 patients randomized in five groups: four groups with different dosages of the drug (2.5, 10, 20, and 40 mg) and a placebo group [47]. The study population included treatment-naïve or background therapy with bosentan (53% of enrolled). After a duration of 16 weeks, the distance walked in 6MWD increased in a dose-dependent way, but only the 40 mg tadalafil group reached the value of statistical significance (*p* < 0.01). Specifically, the improvement in exercise capacity with the highest dosage of tadalafil was higher in naïve patients (44 m) than in the population with background bosentan therapy (23 m), although this was not statistically significant. The incidence of clinical worsening, a secondary endpoint, was decreased in the 40 mg tadalafil group versus placebo.

Following the previous studies, the EVALUATION study (Efficacy and Safety of Vardenafil in the Treatment of Pulmonary Arterial Hypertension) aimed to assess the efficacy and safety of vardenafil in patients with pulmonary hypertension [48]. Sixty-six invited patients were randomly divided in two groups in the ratio 2:1, vardenafil vs. placebo, for a duration of 12 weeks. After 12 weeks, all of the study population was administered 5 mg of vardenafil twice daily for a further 12 weeks. The mean 6MWD, the primary outcome, was significant increased (69 m; *p* < 0.001), and this result was confirmed at the end of the entire study (24 weeks). Concerning hemodynamic outcomes, the administration of vardenafil caused a reduction in pulmonary vascular resistance (24.7 Wood U, *p* < 0.003) and mean pulmonary arterial pressure (25.3 mm Hg, *p* < 0.047) and was associated with an improvement in cardiac index (0.39 L min/m^2^; *p* = 0.005). However, one of the main limitations of PD5i is its dependence on endogenous NO levels.

### 6.2. Soluble Guanylate Cyclase-Stimulator (sGC)

As already stated, soluble guanylate cyclase catalyzes the conversion of GTP into cGMP and is made up of two subunits, one α and one β, that bind heme. Through a dual mechanism of action, sGC stimulators directly target sGC and promote its complete activation, increasing the synthesis of cGMP. Given that they have a unique binding location on sGC, they activate sGC independently of NO. By stabilizing NO–sGC interaction, they may additionally render sGC more sensitive to low levels of NO [49].

Riociguat (BAY 63-2521) is a soluble guanylate cyclase stimulator, and its activity is not limited by low NO endogenous levels, which tends towards low levels as PAH advances [50]. It enhances cGMP levels via two separate mechanisms: first, it stabilizes NO binding to sGC, and second, it directly stimulates sGC independently of NO levels (Figure 4). In the pulmonary circulation, this leads to vascular smooth muscle cell relaxation [51]. That is the reason why, in the past few years, there has been an increasing interest in riociguat, which has been tested in PAH patients. Table 1 shows phase I, II, and III studies with riociguat in pulmonary arterial hypertension and chronic thromboembolic pulmonary hypertension. Both phase I [52] and phase II studies [53] have shown promising results; hence, riociguat was tested in a multicenter, double-blind, randomized controlled trial: the PATENT-1 study (Pulmonary Arterial hyperTENsion sGC-stimulator Trial) [54]. The aim of the study was to evaluate the efficacy of riociguat in the treatment of patients with PAH who are treatment-naïve or on endothelin-receptor antagonists or prostanoids (oral, inhalative, or subcutaneous). The primary endpoint was a difference in 6MWD from baseline after 12 weeks of treatment. Secondary endpoints were pulmonary vascular resistance (PVR), serum levels of NT-proBNP peptide, WHO functional class, time to clinical worsening, Borg scale, quality of life, and safety. A total of 443 patients were randomized in three different groups: placebo (P), riociguat up to 2.5 mg three times daily (R2.5), and riociguat up to 1.5 mg three times daily. The results show a statistically significant difference between the 2.5 mg riociguat group vs. placebo in 6MWD (least-squares mean difference 36 m, 95% CI 20 to 52; *p* < 0.001). Furthermore, there were significant improvements in all secondary endpoints: pulmonary vascular resistance (*p* < 0.001), NT-proBNP levels (*p* < 0.001), WHO functional class (*p* = 0.003), time to clinical worsening (*p* = 0.005), and Borg dyspnea scale (*p* = 0.002). After 12 weeks of follow up, 398 patients were included in the long-term extension study (PATENT-2). Rubin et al. conducted an exploratory analysis of the first 12 weeks of PATENT-2 and found further improvement in the 6MWD of the subgroup of 215 patients who received up to 2.5 mg of riociguat three times daily [55].

Another approved use of riociguat is in the treatment of chronic thromboembolic pulmonary hypertension (CTEPH), a clinical condition characterized by the presence of thrombi in the pulmonary artery. Riociguat was tested in this specific clinical condition and the results were encouraging. First-line therapy in patients affected by CTEPH is pulmonary endarterectomy (PEA), leading to an improvement in symptoms and survival rates [64].

Unfortunately, PEA is not indicated for many patients because of the presence of distal/peripheral CTEPH or severe concomitant illness. Moreover, several patients suffer persistent or recurrent pulmonary hypertension even after pulmonary endarterectomy [65,66]. To address this issue, a specific trial has been designed: the Chronic Thromboembolic Pulmonary Hypertension Soluble Guanylate Cyclase–Stimulator Trial 1 (CHEST), a multicenter phase III study that evaluated the efficacy and safety of riociguat in patients with CTEPH. This was the first study that demonstrated the beneficial effect of pharmacological therapy in CTEPH [56]. The study population was composed of 261 patients with inoperable CTEPH or patients with persistent or recurrent pulmonary hypertension after PEA. They were randomized to a placebo or riociguat group. Patients of the riociguat group demonstrated a statistically significant improvement in 6MWD (primary endpoint of the study) and in PVR, NT-proBNP serum values, and WHO functional class (secondary endpoints) [56]. After 12 weeks, 237 patients entered the long-term extension study (CHEST-2), where assignments were concealed for the first 8 weeks and treatment was open-label thereafter. An exploratory analysis of the first 12 weeks of CHEST-2 showed further improvement in the 6MWD of the group that was already receiving riociguat in the CHEST-1 study [57].

Riociguat benefits were also demonstrated by improvements in the echocardiography parameters of patients with PAH and CTEPH. Importantly, right heart function and size are prognostic factors in PAH and CTEPH, so any effective treatment for PAH or CTEPH must improve the functional and structural parameters of RV function. In 2015, Marra et al. [67] tested riociguat treatment with doses of 1.0–2.5 mg three times daily in 39 patients who were previously enrolled in the trials PATENT, PATENTplus, EAS, and CHEST and continued treatment for 3–12 months. Echocardiography and 6MWD were analyzed at baseline and after 3, 6, and 12 months. Right heart catheterization was performed at baseline and after 3 months. The mean right ventricular (RV) area and right atrial area significantly decreased, respectively, after 3 and 12 months of treatment, while the tricuspid annular plane systolic excursion (TAPSE) significantly improved after 6 months. Furthermore, RV wall thickness and 6MWD significantly improved after 3 months (*p* < 0.05). Invasive hemodynamics significantly improved after 3 months. These results were confirmed by the RIVER study [58], an observational study with the aim of analyzing echocardiographic changes in patients affected by PAH and CTEPH and treated with 1.0–2.5 mg of riociguat three times daily for 3–12 months. The study demonstrated that patients with PAH and CTEPH under long-term treatment with riociguat significantly show improvement in TAPSE, fractional area change, tricuspid regurgitation velocity, and qualitatively assessed RV systolic function. Moreover, the effect of riociguat on RV contractile function was assessed using speckle-tracking echocardiography in 27 patients with PH (26% PAH, 74% CTEPH). Riociguat significantly improved RV systolic function, including RV fractional area change and RV global longitudinal strain [68].

Another class of molecule that plays a role in the NO pathway is the phosphodiesterase type 5 inhibitor (PDE5i), which increase cGMP levels by inhibiting its breakdown.

Riociguat and PDE5i act on the same pathway, but they have an additive vasodilatory effect when used in combination. This is supported by data from the PATENT PLUS study, in which the combination of riociguat and sildenafil in patients with PAH was associated with a reduction in standing systolic blood pressure compared with patients treated with only riociguat [61]. For this reason, and for the lack of substantial benefits from the combination of both riociguat and PDE5i, the combination is contraindicated.

The RESPITE trial [62] analyzed the safety, feasibility, and benefits of switching from PDE5i to riociguat in patients with PAH that have experienced no benefits from PDE5i treatment. RESPITE was a 24-week, open-label, multicenter, uncontrolled study including 61 patients. Exploratory endpoints included change in 6MWD, WHO FC, NT-proBNP, and safety. At the end of observation, the 6MWD had increased by 31 ± 63 m, NT-proBNP decreased by 347 ± 1235 pg·mL^−1^, and the WHO FC improved in 28 patients (54%). A total of 50% of patients experienced side effects and 10% experienced clinical worsening, including death in two patients.

Another important study is the REPLACE trial [63], which assessed the effects of switching to riociguat from PDE5i therapy versus continued PDE5i therapy in patients with PAH. The study included patients with symptomatic PAH at intermediate risk of 1-year mortality, and they were randomized one-to-one to either remain on treatment with PDF5i or switch to 2.5 mg of riociguat three times daily. The endpoint is a composite of absence of clinical complications and predetermined improvements in at least two of three variables (6MWD, WHO functional class, and NT-proBNP). The primary endpoint was met by 45 of 111 (41%) patients in the riociguat group and 23 of 113 (20%) patients in the PDE5i group (odds ratio (OR) 2·78 (95% CI 1.53–5.06; *p* = 0.0007)). Clinical complications occurred in one of 111 (1%) patients in the riociguat group (hospitalization due to aggravate PAH) and 10 of 114 (9%) patients in the PDE5i group (hospitalization due to aggravate PAH [*n* = 9]; disease progression [*n* = 1]; OR 0.10 [0.01–0.73]; *p* = 0.0047).

In summary, all these studies have demonstrated that riociguat significantly improved exercise capacity and secondary efficacy endpoints in patients with pulmonary arterial hypertension. The combination of both riociguat and PDE5i is contraindicated and switching to riociguat from PDE5i treatment could be a strategic option for treatment escalation.

## 7. Conclusions and Future Directions

The nitric oxide pathway has a ubiquitous and central role in the body, having various organs and systems as its target organs. Being a fine regulator of both vascular tone and cell proliferation in the endothelium, it plays a central role in the pathogenesis of PAH. Nowadays, two classes of drugs acting on this pathway are approved for the treatment of pulmonary arterial hypertension: the PDE5i (sildenafil and tadalafil) and the sGC activator (riociguat). While in PAH the PDE5i are first-line drugs and riociguat is a therapy to be implemented only in case of failure of the former, the latter drug turns out to be the only oral medical therapy approved for CTEPH. Taking into consideration all the evidence reported in the current article, drugs modulating the NO pathway play a central role in PAH and CTEPH. However, before using these medications for pulmonary hypertension resulting from other conditions, especially those caused by left heart disease, there are extremes to take into account. An initial trial was conducted in 2012 [69] on 201 patients with pulmonary hypertension caused by systolic left ventricular dysfunction. Although no improvement in the chosen primary endpoint (mean pulmonary arterial pressure) was found in this study, there was significant improvement in some secondary endpoints such as pulmonary vascular resistances (−46.6 dynes-s^−1^·cm^−5^; 95% confidence interval, −89.4 to −3.8; *p* = 0.03) and systemic vascular resistance (−239.3 dynes-s^−1^·cm^−5^; 95% confidence interval, −363.4 to −115.3; *p* = 0.0002) and a relevant improvement in the Minnesota Living With Heart Failure score (*p* = 0.0002). Moreover, a recent study [70] evaluated the effects of riociguat in patients with heart failure with preserved ejection fraction (DYNAMIC study). In this study, there was a significant increase in the primary endpoint, cardiac output (CO), which increased by 0.37 ± 1.263 L/min in the riociguat group and decreased by −0.11 ± 0.921 L/min in the placebo group (*p* = 0.0142), in spite of a significative increase in drop-out patients. Interestingly, a phase III study with a riociguat analog (vericiguat) conducted in patients with heart failure with reduced ejection fraction (HFrEF) without pulmonary hemodynamic characterization reported a significant improvement in the vericiguat-treated group in the composite of heart failure hospitalizations and mortality (primary endpoint) (hazard ratio, 0.90; 95% CI, 0.83 to 0.98; *p* = 0.02) [71].

In conclusion, it is reasonable to consider targeting the nitric oxide pathway, a mainstay in the treatment of PAH and CTPEH with interesting prospects in the field of heart failure as well. This leads to the suggestion to test the NO pathway as a target in other cardiovascular conditions in the future. One of the major limitations in modulating the cGMP signal perhaps lies in our limited understanding of the compartmentalization of cGMP and its functional implications. Solving this limitation could lead to the development of novel drugs that target not only a singular step of the cascade, as riociguat does, for example, but also a particular subcellular compartment where cGMP acts.

## Figures and Tables

**Figure 1 ijms-25-00036-f001:**
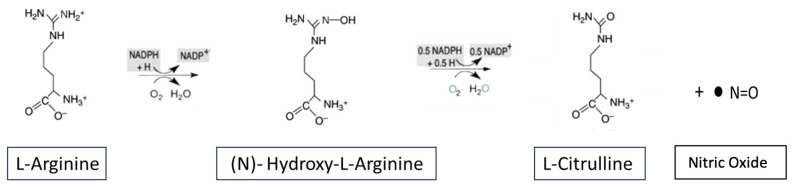
NO Synthesis. Metabolic pathway of NO synthesis from arginine by endothelial, neuronal, or inducible isoforms of the enzyme NO synthase with concomitant formation of citrulline.

**Figure 2 ijms-25-00036-f002:**
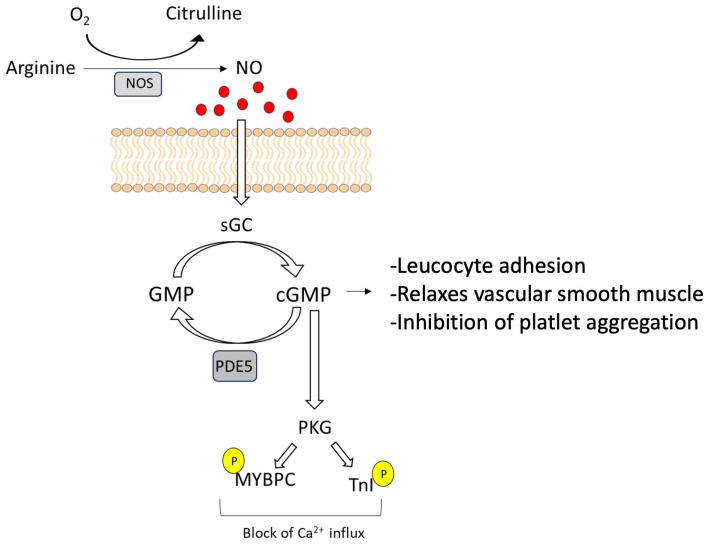
NO effects on the healthy myocardium. NOS–NO signaling effects on cardiovascular function are mediated through direct phosphorylation of myosin-binding protein C (MYBPC) and troponin I (TnI), which are responsible for the reduction in myofilament Ca^2+^ sensitivity.

**Figure 3 ijms-25-00036-f003:**
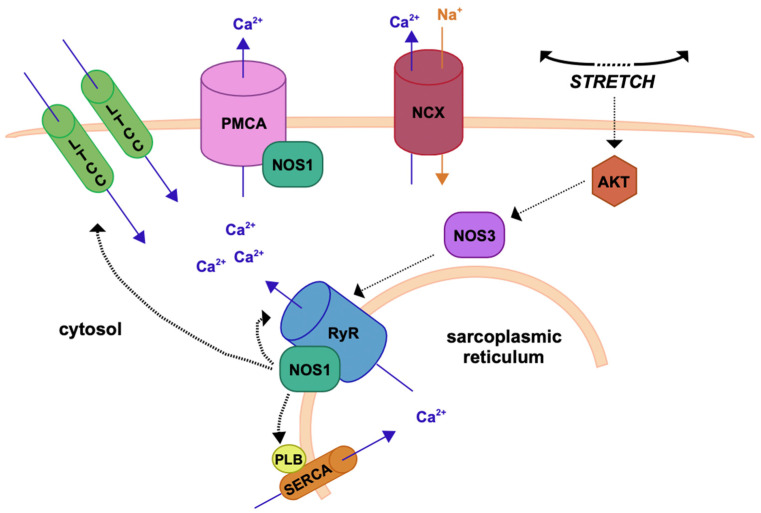
The NO pathway in cardiac myocytes. A schematic illustration of Ca^2+^ fluxes during excitation-contraction in ventricular cardiomyocytes. This diagram depicts the most representative protein complexes and intercellular organelles involved in the cardiac excitation-contraction coupling. Efficient systolic contraction and diastolic relaxation is reliant on efficient Ca^2+^ handling through several processes. Diffusion of Ca^2+^ into the cytosol through LTCC, L-type Ca^2+^ channel; Ca^2+^-induced-Ca^2+^-release from the ryanodine receptors (RyRs); Sequestration of Ca^2+^ back in to the sarcoplasmic reticulum via the important Ca^2+^ pump sarco/endoplasmic reticulum Ca^2+^ ATPase (SERCA); and Expulsion of Ca^2+^ from the cell via sodium-calcium exchanger pumps (NCX); PLB, phopholamban; Ca^2+^, calcium.

**Figure 4 ijms-25-00036-f004:**
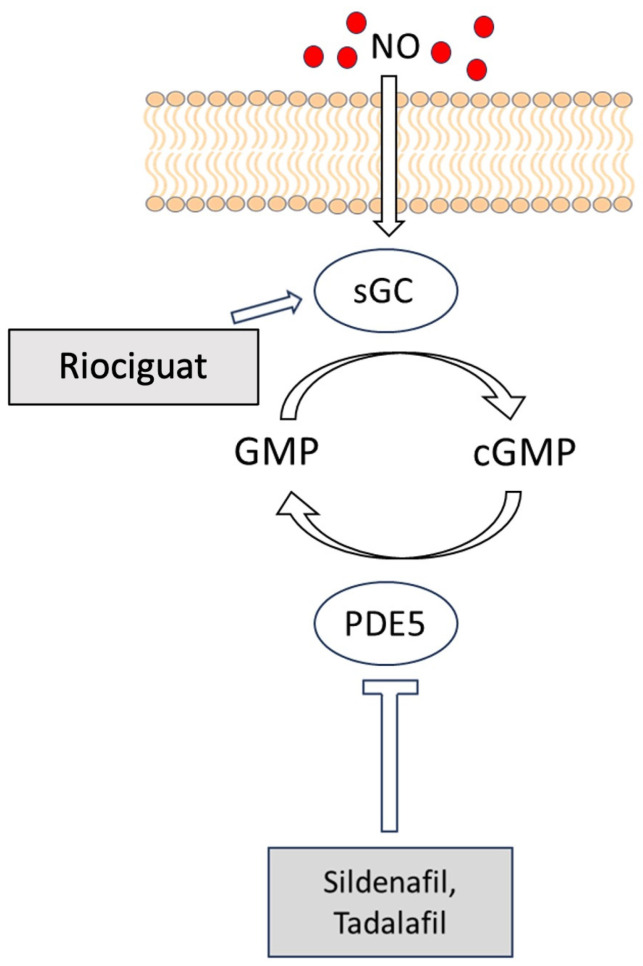
Mode of action of riociguat and PDE5i. Riociguat acts on the nitric oxide (NO) receptor, soluble guanylate cyclase (sGC), and stimulates the enzyme independently of NO. On the other side, PDE5i bind to the catalytic site of the PDE5 enzyme to act as a competitive inhibitor of cGMP, avoiding its conversion to GMP.

**Table 1 ijms-25-00036-t001:** Phase I, II, and III studies with riociguat in pulmonary arterial hypertension and chronic thromboembolic pulmonary hypertension.

Studies	Sample	Drug Used	Main Results
“First acute haemodynamic study of soluble guanylate cyclase stimulator riociguat in pulmonary hypertension” Grimminger et al., 2009 [52]	PAH *n* =19	Riociguat 2.5 or 1 mg	Improving of pulmonary hemodynamic parameters and cardiac index in patients with PH
“Riociguat for chronic thromboembolic pulmonary hypertension and pulmonary arterial hypertension: a phase II study” Grophani et al., 2010 [53]	PAH *n* = 33; CTEPH *n* = 42	Riociguat 1.0–2.5 mg three times daily	Increasing of 6MWD in both CTEPH and PAH; reduction in pulmonary vascular resistance; asymptomatic hypotension (SBP <90 mmHg) normalized after dose reduction in two patients. Dyspepsia, headache, and hypotension
“Riociguat for the treatment of pulmonary arterial hypertension” Grophani et al., 2013 [54]	PAH *n* = 443	Riociguat 2.5 mg three times daily	Increasing of 6MWD (*p* < 0.001); significant improvements in pulmonary vascular resistance (*p* < 0.001); NT-proBNP levels (*p* < 0.001); WHO functional class (*p* = 0.003); time to clinical worsening (*p* = 0.005); and Borg dyspnea score (*p* = 0.002). Syncope in four patients
“Riociguat for the treatment of chronic thromboembolic pulmonary hypertension: a long-term extension study (CHEST-2)” Ghofrani et al., 2013 [56]	PAH *n* = 261	Riociguat 0.5–2.5 mg three times daily	Statistically significant (*p* < 0.001); reduction in pulmonary vascular resistance (*p* < 0.001); improvements in the NT-proBNP level (*p* < 0.001); WHO functional class (*p* = 0.003); right ventricular failure (3% of patients) and syncope (2% of patients)
“Riociguat for the treatment of chronic thromboembolic pulmonary hypertension: a long-term extension study (CHEST-2)” Simmoneau et al., 2015 [57]	PAH *n* = 243	Riociguat 2.5 mg three times daily for 12 weeks	Further improvement in the 6MWD after long-term riociguat administration
“Riociguat for the treatment of pulmonary arterial hypertension: a long-term extension study (PATENT-2)” Rubin et al., 2015 [55]	PAH *n* = 324	Riociguat 1.0–2.5 mg three times daily	Further improvement in the 6MWD after long-term riociguat administration of 2.5 mg t.i.d.
“Right ventricular size and function under riociguat in pulmonary arterial hypertension and chronic thromboembolic pulmonary hypertension (the RIVER study)” Marra AM et al., 2018 [58]	PAH *n* = 32; CTEPH *n* = 39	Riociguat 1.0–2.5 mg three times daily	Significant reduction in RA and RV area and RV thickness tricuspid regurgitation velocity; TAPSE improvement; RV thickness and RV fractional area change
“Clinical Significance of Guanylate Cyclase Stimulator, Riociguat, on Right Ventricular Functional Improvement in Patients with Pulmonary Hypertension” Murata M et al., 2021 [59] “Riociguat and the right ventricle in pulmonary arterial hypertension and chronic thromboembolic pulmonary hypertension” Benza et al., 2022 [51,59]	PAH *n* =14; CTEPH *n* = 31	Riociguat mean final daily dose of 7.2 ± 0.9 mg administered for a mean 234 days	Improving RV global longitudinal strain
“Right ventricular size and function under riociguat in pulmonary arterial hypertension and chronic thromboembolic pulmonary hypertension (the RIVER study)” Marra AM, 2018 [58] “Riociguat and the right ventricle in pulmonary arterial hypertension and chronic thromboembolic pulmonary hypertension” Benza et al., 2022 [51]	PAH *n* = 32; CTEPH *n* = 39	Riociguat 1.0–2.5 mg three times daily for 3–12 months	Improving RV systolic function
“Riociguat, a soluble guanylate cyclase stimulator, ameliorates right ventricular contraction in pulmonary arterial hypertension” Murata et al., 2018 [60] “Riociguat and the right ventricle in pulmonary arterial hypertension and chronic thromboembolic pulmonary hypertension” Benza et al., 2022 [51]	PAH *n* = 7; CTEPH *n* = 20	Riociguat mean dose of 7.3 ± 0.7 mg administered for a mean 220 days	Increased RV fractional area
“PATENT PLUS: a blinded, randomised and extension study of riociguat plus sildenafil in pulmonary arterial hypertension” N. Galiè et al., 2015 [61]	PAH *n* = 12	Sildenafil 20 mg t.i.d. and randomization to placebo or riociguat up to 2.5 mg three times daily for 12 weeks	Potentially unfavourable safety signals with sildenafil plus riociguat and no evidence of a positive benefit/risk ratio
“RESPITE: switching to riociguat in pulmonary arterial hypertension patients with inadequate response to phosphodiesterase-5 inhibitors” Hoeper et al., 2017 [62]	PAH *n* = 51	1–3 day PDE5i treatment-free period before receiving riociguat adjusted up to 2.5 mg maximum three times daily for 24 weeks	Improvement in 6MWD; reduction in NT-proBNP; improvement in World Health Organization (WHO) functional class; 52% of patients experienced study-drug-related adverse events (none during the PDE5i treatment-free period). Six patients experienced clinical worsening, including death in two
“Switching to riociguat versus maintenance therapy with phosphodiesterase-5 inhibitors in patients with pulmonary arterial hypertension (REPLACE): a multicentre, open-label, randomised controlled trial” Hoeper et al., 2021 [63]	PAH *n* = 211	Oral PDF5-I (sildenafil ≥60 mg or tadafanil 20–40 mg) or switching to oral riociguat up to 2.5 mg three times daily	Improvements in at least two of three variables (6MWD, WHO functional class, and NT-pro BNP) was met in 41% of patients treated with riociguat and in 20% of patients treated with PDF5i

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
