# Peer review of "Modulating NO–GC Pathway in Pulmonary Arterial Hypertension"

_ijms, 2023, doi:10.3390/ijms25010036_

Round 1

Reviewer 1 Report

Comments and Suggestions for Authors

Comments in the attached document

Comments on the Quality of English Language

Minor revisions

Author Response

Dear Reviewer,
we thank you for your efforts and time spent in reviewing our manuscript. Please find attached our responses.

Reviewer 2 Report

Comments and Suggestions for Authors

Overall, this comprehensive review article is well-written and presents a novel concept for reviewing the idea of modulating the nitric oxide (NO)-guanylate cyclase (GC) pathway for Pulmonary Arterial Hypertension (PAH). PAH is a progressive disease characterized by increased pulmonary vascular resistance and pulmonary arterial pressure, which can lead to right heart failure and death. The NO-GC pathway is a key regulator of vascular tone and homeostasis, and its dysfunction has been implicated in the development of PAH. The authors discuss the potential of modulating the NO-GC pathway through various approaches, including sGC stimulators and PDE5 inhibitors. The authors conclude that modulating the NO-GC pathway has shown promise in both preclinical and clinical studies and may offer a novel therapeutic approach for PAH. Several concerns are outlined below. 

1. I recommend that the authors refer to the 2022 ESC/ERS Guidelines for the diagnosis and treatment of pulmonary hypertension. Additionally, I suggest consulting recent articles, such as those published in 2023.

 2. Spelling issue on line 83; 'manly' should be corrected to 'mainly'. Also, line 190, 'ha'. 

3. In lines 88-91, the sentence 'It has been...' is using the same wording as the reference. The authors should rewrite the sentence using their own phrasing.

 4. The authors should confirm whether the statement in line 105-107, 'cGMP as a second messenger is responsible for a variety of downstream effects such as leucocyte adhesion, relaxation of vascular smooth muscle, and inhibition of platelet aggregation' is referenced as [21]. 

5. At line 246, the PACES study (…) should include the closing parenthesis. 

6. Ensure consistent formatting for abbreviations, for example, ‘TID, t.i.d’, ‘6MWD, 6-MWD’, ‘P, p’. 

7. In subtitle 7, 'Modulation of Pulmonary Vascular NO in PAH,' the authors mention two classes of drugs that modulate the NO/cGMP pathway, including PDE5 inhibitors and sGC. However, in the subtitle 7, only the PDE5 inhibitor is covered, while sGC is introduced in subtitle 8. Perhaps it would be more appropriate to include both PDE5 inhibitors and sGC in subtitle 7.

 8. The second reference is using a different citation format. 

9. The review article needs to be thoroughly reviewed by the authors, as there are spelling mistakes.

Author Response

(The authors gave the same response as above.)

Round 2

Reviewer 1 Report

Comments and Suggestions for Authors

A very interesting review of the regulation of the NO-cGMP-GC pathway in the cardiovascular regulation associated with the treatment of PAH.

Minor comments:

Line 104 : VSMC is not defined, but it is line 138. It is defined again unnecessarily on line 155

Line 157 : neuronal nitric oxide synthase (nNOS) and line 69: neurological NOS (nNOS). Do not define it twice and neurological is wrong

Line 167: Ca2+ intracellular levels => intracellular Ca2+ levels

Line 174 and 176: Ca2+ => Ca2+

Line 185: O2- => O2-

Line 212: through regulation of excitation–contraction => through regulation of excitation–contraction coupling.

Line 199: that needs to be considerd => that needs to be considered

Line 272: WHO functional class at the end of 12 weeks=> World Health Organization (WHO) functional class at the end of 12 weeks

Line 341: World Health Organization (WHO) functional class => WHO functional class

Reviewer 2 Report

Comments and Suggestions for Authors

Authors have answered all my requests.

Author Response

Thank you very much for taking the time to review this manuscript. We believe that your efforts have significantly improved our work, and for that we are grateful.